# Bismuth-Rich Co/Ni Bimetallic Metal–Organic Frameworks as Photocatalysts toward Efficient Removal of Organic Contaminants under Environmental Conditions

**DOI:** 10.3390/mi14050899

**Published:** 2023-04-22

**Authors:** Ayesha Siddiqa, Toheed Akhter, Muhammad Faheem, Shumaila Razzaque, Asif Mahmood, Waheed Al-Masry, Sohail Nadeem, Sadaf Ul Hassan, Hyunseung Yang, Chan Ho Park

**Affiliations:** 1Department of Chemistry, School of Science, University of Management and Technology, C-II, Johar Town, Lahore 54770, Pakistan; 2Department of Chemical Engineering, College of Engineering, King Saud University, Riyadh 11421, Saudi Arabia; 3Department of Chemistry, COMSATS University Islamabad, Lahore Campus, Lahore 54000, Pakistan; 4Electronic Convergence Materials & Device Research Center, Korea Electronics Technology Institute, Seongnam-si 13509, Republic of Korea; hsyang@keti.re.kr; 5Department of Chemical and Biological Engineering, Gachon University, 1342 Seongnam-daero, Seongnam-si 13120, Republic of Korea

**Keywords:** MOF, composites, heterostructures, photocatalysis, organic pollutant

## Abstract

Active photocatalysts with an efficiency of 99% were prepared for the degradation of the industrial dye, methylene blue (MB), under visible light irradiation. These photocatalysts comprised Co/Ni-metal–organic frameworks (MOFs), to which bismuth oxyiodide (BiOI) was added as a filler to prepare Co/Ni-MOF@BiOI composites. The composites exhibited remarkable photocatalytic degradation of MB in aqueous solutions. The effects of various parameters, including the pH, reaction time, catalyst dose, and MB concentration, on the photocatalytic activity of the prepared catalysts were also evaluated. We believe that these composites are promising photocatalysts for the removal of MB from aqueous solutions under visible light.

## 1. Introduction

Extensive industrial manufacturing results in the uncontrolled release of toxic effluents, mainly organic dyes, into natural water resources. These effluents cause various hazardous environmental problems that endanger living species [1]. The textile, paper, printing, and paint industries are the primary sources of environmental dye pollution, and azo dyes contribute to 50% of the global dye production [2]. Therefore, an efficient method is required to remove organic contaminants from water resources and the environment. To date, various methods have been adopted, including adsorption, advanced oxidation processes, membrane separation, coagulation, and photocatalytic degradation of the pollutants [3]. Among these, the use of catalysts to remove toxic organic contaminants from air and water is considered an environmentally friendly technique that does not cause secondary pollution [4]. This method uses safe, clean, and renewable solar energy sources to alleviate pollution. Consequently, photocatalysis has attracted considerable attention in recent years [5]. In this regard, various metal oxide- and metal sulfide-based photocatalysts such as TiO_2_ [6], ZnO [7], WO_3_ [8], ZnS [9], and CuO [10] have been extensively reported. Owing to their large bandgaps, these photocatalysts can respond to ultraviolet or near-ultraviolet radiation, which limits their potential for photocatalytic degradation of organic pollutants. Therefore, photocatalysts modified to absorb visible light are required [11]. In this regard, various photocatalysts with narrow bandgaps and prompt responses to visible light have been reported, including Bi_2_SbVO_7_ [12], CaBi_6_O_10_ [11], ZnGa_2_O_4_ [13], CaBi_2_O_4_ [14], and BiOX (X = Cl, Br, I) [15]. The photocatalytic activity of these materials is attributed to the hybridization between the nd [10] outer layer orbitals of metal ions and the O 2p orbitals, which pushes the valence band upward, thus leading to a narrow bandgap [16]. Among these, BiOI, having a narrow bandgap of 1.7–1.9 eV, shows strong visible light absorption, which affords it good photocatalytic activity [17]. However, charge carrier recombination is a limitation when using BiOI as a single photocatalyst; therefore, BiOI requires further modifications for practical use. To address this problem, BiOI-based heterostructures have been constructed to prevent the recombination of photoexcited charge carriers [18]. Recently, various BiOI-based heterostructures have exhibited significant photocatalytic activity for the degradation of organic pollutants; for example, Fe_3_O_4_/BiOI hybrids [19], graphene oxide-BiOI hybrids [20], and TiO_2_@BiOI hybrids [21]. Therefore, tailoring heterostructures in which BiOI is coupled with other compounds is an important strategy for improving the performance of photocatalysts. Metal–organic frameworks (MOFs) exhibit good potential as photocatalysts because of their metal–ligand coordination, suitable porous structure, and high surface area [22]. Previously, MOFs such as [Ni_2_(C_10_H_8_N_2_)_2_][C_12_H_8_O(COO)_2_]_2_⋅H_2_O,[Co_2(_C_10_H_8_N_2_)][C_12_H_8_O(COO)_2_]_2_ [23], [Co(4,4′-bipy)⋅(HCOO)_2_]n, and [Cu(4,4′-bipy)Cl]n [24] (comprising a single metal) have displayed notable photocatalytic degradation of organic pollutants. Later, it was revealed that MOFs consisting of more than one metal can perform better in photocatalytic applications, which was attributed to the synergistic effect of the metals [25]. It was also shown that bimetallic MOFs, wherein two metals are coordinated with an organic ligand, afforded more tenability in structural design through the facile variation of the ratio of the two different metals [26]. The degradation of organic-based industrial pollutants was investigated using different bimetallic MOFs, including FeNiX-BDC [27] and Co/Ni-MOF-74 [26]. However, the low absorption of visible light by bimetallic MOFs limits their potential. Heterostructures prepared by combining two photocatalysts have been used to enhance the photocatalytic degradation of organic contaminants [4].

Recently, improved photocatalytic activity has been achieved by mixing BiOI with reduced GO, MOFs, and other materials. The enhanced activity has been attributed to better visible light absorption, enhanced surface area, and reduced recombination of electron–hole pairs [20,28]. Several investigations have been conducted to construct heterostructures such as BiOI with MOF, rGO, and g-C3N4 to enhance visible light absorption, increase the surface area, and inhibit the recombination of photoexcited electron–hole pairs. However, the incorporation of BiOI into the bimetallic Co/Ni-2,5-dihydroxyterephthalic acid (DHTA) MOF to prepare heterostructures for the efficient photocatalytic degradation of organic pollutants under visible light has not yet been explored.

In this study, we synthesized BiOI-containing bimetallic MOFs for photocatalytic removal of organic pollutants as shown in Figure 1. Ni-MOF was synthesized using a single-step solvothermal method in the first step; Co/Ni-MOF was prepared by doping Ni-MOF with Co in the second step. Subsequently, BiOI was incorporated into the Co/Ni-MOF to prepare a heterostructure, which was denoted as Co/Ni-MOF@BiOI. UV–visible spectroscopy, X-ray diffraction (XRD), and scanning electron microscopy (SEM) were used to thoroughly analyze the prepared materials. The photocatalytic activity of the Co/Ni-MOF@BiOI composites was assessed based on the degradation of methylene blue (MB) under visible light irradiation in the presence of the composites. The effects of reaction parameters such as catalyst dosage, dye concentration, and pH on the photocatalytic performance of the prepared catalysts were also examined. These performance evaluation studies revealed that the Co/Ni-MOF@BiOI composites have excellent catalytic efficiency for the photodegradation of MB under visible light illumination.

## 2. Experimental

### 2.1. Materials

All the chemicals were purchased from Sigma-Aldrich; they were of a high purity and were used without further purification. Nickel nitrate hexahydrate (Ni(NO_3_)_2_·6H_2_O) (98%), Cobalt nitrate hexahydrate (Co(NO_3_)_2_·6H_2_O) (97%), 2,5-dihydroxyterephthalic acid (DHTA) (98%), N,N-dimethylformamide (DMF) (99.5%), bismuth nitrate pentahydrate (Bi(NO_3_)_3_·5H_2_O) (71%), potassium iodide (KI) (99.5%), ethylene glycol (EG) (99%), and ethanol were used.

### 2.2. Synthesis of Ni-MOF

To synthesize the Ni-MOFs, two solutions (A and B) were prepared. Solution A was prepared by dissolving 1.46 g (Ni(NO_3_)_2_·6H_2_O) in 30 mL DMF. Solution B was prepared by dissolving DHTA (0.5 g) in dimethylformamide (DMF, 30 mL) and stirring for 30 min. Solution B was then added to solution A, and 0.3 mL HCl was added dropwise to the mixture as a chemical modulator for crystal growth [29,30]. The resulting green solution containing a metal-to-organic linker molar ratio of 2:1 was poured into a Teflon-lined autoclave and heated at 180 °C for 24 h. After cooling to 25 °C, yellow precipitates formed, were separated by centrifugation, and washed with DMF and ethanol before being dried in a vacuum oven at 60 °C for 12 h.

### 2.3. Synthesis of Co/Ni-MOF

Co-doped Ni-MOF was synthesized using the same method used for Ni-MOF, except for the addition of Co(NO_3_)_2_·6H_2_O. In a typical procedure, 0.74 g (0.002 mol) Co(NO_3_)_2_·6H_2_O and 1.46 g (0.004 mol) Ni(NO_3_)_2_·6H_2_O were dissolved in solution A (molar ratio of Co(NO_3_)_2_·6H_2_O/Ni(NO_3_)_2_·6H_2_O = 0.5). The rest of the procedure was the same as previously described.

### 2.4. Synthesis of Co/Ni-MOF@BiOI

To synthesize the Co/Ni-MOF@BiOI composites, a solution of Bi(NO_3_)_3_·5H_2_O (0.004 mol) in EG (25 mL) was prepared. Co/Ni-MOF (0.06 g) was gradually added to this solution while stirring to form a yellow suspension. KI (0.33 g) was dissolved in EG (25 mL) and added dropwise to the suspension while stirring for 1 h. The resulting yellow solution was poured into a Teflon-lined autoclave and heated at 120 °C for 12 h, forming precipitates. These precipitates were collected via centrifugation, rinsed with distilled water, and dried in a vacuum oven at 60 °C overnight. By varying the molar ratio of Co/Ni-MOF to BiOI, two composites were prepared: Co/Ni-MOF@BiOI-5 (with 5% BiOI by weight) and Co/Ni-MOF@BiOI-10 (with 10% BiOI by weight).

## 3. Characterization

Phase identification and determination of the crystallinity of the as-prepared samples were conducted using an X-ray diffractometer (XRD; Bruker D2) with Cu Kα radiation (λ = 0.15406 nm) at 2θ = 10–80°. The morphologies of the samples were observed using field-emission scanning electron microscopy (FE-SEM; Nova NanoSEM 450). UV–vis spectroscopy (Metash UV-9000) was used to study the optical properties. A JEOL-JEM 2010 field transmission electron microscope (TEM) operating at 200 kV was utilized to obtain TEM and high-resolution transmission electron microscopy (HRTEM) images.

### Photocatalytic Degradation Experiment

The photocatalytic degradation of MB was examined to evaluate the efficiency of the as-prepared photocatalysts under sunlight. A MB solution of 10 ppm concentration was freshly prepared from a stock solution [1000 mg L^−1^]. Photocatalytic activity was investigated in a batch experiment in which adsorption and photocatalytic degradation were simultaneously conducted. The effect of contact time on the removal of cationic MB was investigated using all the prepared photocatalysts (Ni-MOF, Co/Ni-MOF, BiOI, Co/Ni-MOF@BiOI-5, and Co/Ni-MOF@BiOI-10). To achieve adsorption–desorption equilibrium, an aqueous solution of dye (10 ppm) containing 0.08 g photocatalyst was stirred for 30 min in the dark. The dye–photocatalyst mixture was then transferred to a Petri dish and exposed to sunlight. Samples were collected at specific time intervals (0, 60, 120, 180, and 240 min) and centrifuged for 2 min. The residual dye concentration was examined using UV–visible spectroscopy, and the degradation efficiency was determined using the following equation:(1)$$R\;\left(\%\right)=C_{o}-\frac{C_{t}}{C_{o}}$$
where *C_o_* and *C_t_* are the initial and final MB concentrations, respectively. The efficiencies of all the prepared photocatalysts toward the degradation of MB were studied using various parameters such as time interval, pH, dye concentration, and catalyst dosage.

## 4. Results and Discussion

### 4.1. XRD Analysis

The XRD patterns of Ni-MOF, Co/Ni-MOF, BiOI, Co/Ni-MOF@BiOI-5, and Co/Ni-MOF@BiOI-10 were studied to affirm the preparation of these samples, all of which are exhibited in Figure 2. All prepared samples were crystalline, as observed from the XRD patterns. In the diffractogram of Ni-MOF, the diffraction peaks were observed at 2θ = 11.8°, 18.01°, 22.28°, 23.83°, and 33.48°, corresponding to the (111), (150), (220), (024), and (025) [31] planes (JCPDS number = 035-167). Figure 2 shows that the XRD pattern of Co/Ni-MOF resembles that of Ni-MOF, indicating that the addition of the doped metal did not disturb the structural framework of the single-metal MOF (Ni-MOF) [32].

The XRD pattern of BiOI displayed diffraction peaks at 2θ = 26°, 32.7°, 46.9°, and 54.7°, which can be assigned to the (102), (110), (200), and (212) planes [33]. There were no extra peaks in the XRD patterns of the composites, indicating the crystallinity and purity of Co/Ni-MOF@BiOI. Additionally, the crystallite size of Co/Ni-MOF@BiOI reduced to 10.58 nm in comparison to Co/Ni-MOF, where it was 23.93 nm, as calculated using the Scherrer equation (D = 0.9λ/βcosθ) (Appendix A). Hence, a reduction in the crystallite size results in a larger surface area, which provides an excellent platform for wide interactions with organic pollutants in photocatalytic applications. The diffraction peaks of BiOI appeared more prominently in the Co/Ni-MOF@BiOI composites, which can be attributed to their smaller particle size and/or larger lattice strain [29]. The XRD patterns of all the prepared samples were in good agreement with the literature [31,33].

### 4.2. SEM Analysis

The microstructures and surface morphologies of all the prepared samples were analyzed using SEM. As shown in Figure 3, Ni-MOF has a layered structure with a rod-like morphology. In the case of Co/Ni-MOF, the nanorod shape was retained; however, the average diameter (n > 100) expanded to 87.5 nm. These nanorods are interlaced to form a flower-like pattern, as shown in Figure 3c,d. Furthermore, an HR-TEM image of Co/Ni-MOF is presented in Appendix A. HR-TEM further confirmed the crystalline nature of Co/Ni-MOF and the continuous elemental distribution of Co, Ni in the sample. This TEM image also reveals the position of different elements in the electrocatalyst.

Similarly, the prepared BiOI comprised nanopetals that combined to form flower-like shapes (Appendix A). In addition, the EDX spectrum of Ni-MOF showed the presence of Ni in the sample, whereas both Co and Ni were detected in the EDX spectrum of Co/Ni-MOF (Appendix A). Figure 4 shows the morphologies of Co/Ni-MOF@BiOI-5 and Co/Ni-MOF@BiOI-10. A rod-like morphology was also observed in the Co/Ni-MOF@BiOI heterostructures. Moreover, these rods were densely packed to form flower-like structures. The average diameter of the rods was found to be 121.3 nm. The increased size of the Co/Ni-MOF@BiOI heterostructure rods over the MOF rods can be attributed to the anchoring of MOF and BiOI to one another, as reported elsewhere [4]. Furthermore, the presence of Bi, I, Co, and Ni in the Co/Ni-MOF@BiOI heterostructures is confirmed by the EDX spectrum shown in Appendix A. The amount of Co/Ni-MOF nanorods increased with the increasing MOF-to-BiOI ratio. In the BiOI/MOF composite, the nanorods became wider (121.3 nm), which was beneficial for the formation of a 3D flower-like shape that provided a more open hierarchical structure and easy access to the active sites [34], thus helping to enhance the photocatalytic activity of the Co/Ni-MOF@BiOI composites.

### 4.3. UV–Visible Spectroscopy

The optical properties of all the prepared samples were studied using UV–visible spectroscopy to understand the reasons for their different photocatalytic activities. Figure 5 shows the UV–visible spectra of Co/Ni-MOF and Co/Ni-MOF@BiOI-10 as representative spectra, and the spectra of all the other samples are provided in the Appendix A.

As shown in Figure 5a, both samples display two absorption bands at 200 and 300 nm, which can be associated with the organic aromatic C_6_ ring [35]. These bands can be assigned to π(HOMO)→ π*(LUMO) transitions due to ^1^A_1g_→^1^B_1u_ and ^1^A_1g_→^1^B_2u_ excitations, respectively [36]. Both samples displayed significant visible light absorption. However, their absorption edges appeared at different wavelengths: 522 nm for Cc/Ni-MOF and 596 nm for Co/Ni-MOF@BiOI-10. This red shift in the absorption edge of Co/Ni-MOF@BiOI-10 compared to that of Co-Ni-MOF indicates that Co/Ni-MOF shows better photocatalytic activity toward the degradation of organic dyes. These results are in good agreement with previously reported data [37].

### 4.4. Photocatalytic Activity

The catalytic performance of Ni-MOF, Co/Ni-MOF, and their composites was investigated by studying the degradation of MB under sunlight, as shown in Figure 6a–c. For this purpose, a reaction mixture containing an aqueous solution of MB (10 ppm) and a specific amount of the photocatalyst was stirred in the dark for 30 min to attain adsorption equilibrium. This reaction mixture was then exposed to direct sunlight, the details of which are shown in Appendix A. Aliquots were collected after irradiation for 0, 60, 120, 180, and 240 min. These aliquots were then subjected to UV–visible spectroscopy to assess any decrease in the MB concentration.

As shown in Figure 6a, only 18% of the MB was degraded when its aqueous solution (10 ppm) was irradiated with sunlight for 240 min in the presence of 0.08 g Ni-MOF as a photocatalyst at a neutral pH. This relatively low photocatalytic activity of Ni-MOF is due to its low light absorption property in the range of visible light. Moreover, the removal of MB from aqueous solutions was primarily achieved by adsorption on the surface of the photocatalyst. Similarly, Co/Ni-MOF removed MB (from an aqueous solution) via adsorption, with a removal efficiency of 34%. The addition of Co^+2^ ions to Ni-MOF may be responsible for the higher efficiency of Co/Ni-MOF. Based on other studies, doped metals can act as mediators between the linker and metal, enhancing photocatalytic activity [38]. BiOI can remove 49% of MB from the aqueous solution, as exhibited in Figure 6a,b. Co/Ni-MOF@BiOI-5 showed a 58% degradation of MB. The maximum degradation was exhibited by Co/Ni-MOF@BiOI-10 (85%). The higher photocatalytic activity of the composites compared to Ni-MOF and Co/Ni-MOF can be explained by the fact that the addition of BiOI to the MOF facilitates the separation of photoexcited electron–hole pairs, leading to increased photocatalytic efficiency [39]. Similarly, among the composites, the increased photocatalytic activity of MOF@BiOI-10 can be attributed to its higher BiOI content compared to that of MOF@BiOI-5. The decrease in the concentration of MB as a function of irradiation time in the presence of each photocatalyst is shown in Figure 6b. Similarly, Figure 6c illustrates MB degradation as a function of irradiation time using Co/Ni-MOF@BiOI-10 as the photocatalyst. It is evident from the figure that the intensity of the peaks in the UV–visible spectra gradually decreases with an increase in irradiation time, which was also visually observed as a change in the color of the solution. Moreover, it can be seen that there is no miscellaneous peak in the spectra, which provides evidence that MB undergoes complete degradation without the formation of any stable by-product [40]. In conclusion, the Ni-MOF and Co/Ni-MOF did not exhibit considerable photocatalytic activity because they were not photoexcited by visible light. In contrast, when BiOI was added to the MOF to prepare the MOF@BiOI heterostructure, a dramatic increase in the photocatalytic activity was observed. The improved photocatalytic efficiency of the Co/Ni-MOF@BiOI composites can be attributed to their higher absorption of visible light and greater separation of the photoexcited electron–hole pairs.

### 4.5. Effect of Various Parameters on MB Degradation

Because Co/Ni-MOF@BiOI-10 showed excellent photocatalytic performance, this sample was chosen to study MB degradation using various parameters such as pH, catalyst dosage, and dye concentration.

### 4.6. Effect of pH

pH is a critical parameter that influences the photocatalytic degradation of organic pollutants. The photocatalytic degradation of organic dyes depends on the pH of the system because any change in pH results in a change in the surface charge, degree of ionization of the dyes, and electrostatic interactions between the photocatalyst and dye [41,42]. The degradation of MB under visible light in the presence of a Co/Ni-MOF@BiOI-10 composite as a function of pH was investigated, and the results are presented in Figure 7a. To evaluate the effect of pH, the degradation of MB was studied at pH values of 3, 5, 7, and 9. HCl and NaOH were used to adjust the pH values. It is obvious from Figure 7a that at pH 5, the photocatalytic efficiency of the Co/Ni-MOF@BiOI-10 composite was remarkably improved, exhibiting a value of approximately 99% toward the degradation of MB under visible light illumination. The catalytic efficiency decreased with increasing pH, displaying values of 97% and 93% at pH 7 and 9, respectively. Thus, the higher degradation efficiency of Co/Ni-MOF@BiOI-10 at pH 5 suggests that mildly acidic conditions are favorable for MB degradation in the presence of a photocatalyst. These findings are in good agreement with those of Fenton reactions conducted under acidic conditions. In the case of photocatalysts, the valence band hole potential decreases with increasing pH value. This weakened the oxidizing ability of the holes, resulting in a reduced rate of hydroxyl radical formation [43].

### 4.7. Effect of Dye Concentration

To evaluate the effects of the initial MB concentration on the photocatalytic efficiency of Co/Ni-MOF@BiOI-10, a 10–30 ppm MB aqueous solution was used. As shown in Figure 7b, the MB degradation decreased with increasing initial concentration. This phenomenon may be ascribed to light-screening effects, which eventually lead to the decreased activity of the photocatalysts. Furthermore, visible light cannot penetrate deep into the solution because of the high concentration of the dye, which limits the availability of light required for photocatalytic activity [44]. Consequently, the photocatalytic activity was negatively affected by high concentrations of MB.

### 4.8. Effect of Catalyst Dosage

Figure 7c,d show the photodegradation performance of Co/Ni-MOF@BiOI-10 toward MB as a function of the catalyst dose. It was observed that the degradation of MB increased with increasing amounts of the photocatalyst until equilibrium was reached. As shown in Figure 7c,d, the efficiency of Co/Ni-MOF@BiOI-10 improved when its concentration was raised from 0.04 to 0.16 g. This indicates that the higher availability of active sites created more photoexcited electron–hole pairs, which resulted in the rapid degradation of the dye [45]. In addition, the photocatalytic performance was saturated at higher catalyst dosages. A high photocatalyst concentration hindered the penetration of visible light, owing to the overlapping or agglomeration of active sites [4].

A comparison of the photodegradation of MB using Co/Ni-MOF@BiOI-10 in the current study and in previously published literature is presented in Table 1. These data indicated that the prepared materials are high-performance photocatalysts that can degrade organic dyes in water under environmental conditions.

### 4.9. Possible Mechanistic Explanation of MB Photodegradation

It has been reported that, when exposed to light, the photoexcited holes (h^+^) in a photocatalyst react with H_2_O to form hydroxyl radicals (·OH) and O_2_, whereas the photoexcited electrons (e^−^) generate superoxide radicals (·O_2_^−^) upon reaction with oxygen [46]. Both hydroxyl and superoxide radicals can oxidize the adsorbed MB into CO_2_. Additionally, the photoexcited e^−^ and h^+^ can either recombine by attacking the MB molecules or react with the dye adsorbed on the photocatalyst surface [46]. According to various reports, superoxide radicals are the most active species for the photodegradation of MB on the surface of a photocatalyst [44].

Based on these reports, the photocatalytic process can be represented as shown below:(2)$$\mathrm{MOF}@\mathrm{BiOI}10+h\nu \;\mathrm{MOF}@\mathrm{BiOI}10\;\left(e^{-}+h^{+}\right)$$
(3)$$\mathrm{MOF}@\mathrm{BiOI}10\left(h^{+}\right)+\mathrm{MB}\;\to \mathrm{Degradation}\;\mathrm{products}$$
(4)$$\mathrm{MOF}@\mathrm{BiOI}10\left(e^{-}\right)+O_{2}\to \;\cdot O_{2}^{-}+\mathrm{MOF}@\mathrm{BiOI}10$$
(5)$$\mathrm{MOF}\left(e^{-}\right)\;\to \mathrm{BiOI}\left(e^{-}\right)$$
(6)$$\mathrm{MB}+\cdot O_{2}^{-}\;\to {\mathrm{CO}}_{2}+{\;H}_{2}O$$

Under visible light irradiation, the Co/Ni-MOF@BiOI-10 catalyst was photoexcited, generating electrons (e^−^) and holes (h^+^) in the conduction and valence bands, respectively, as shown in Equation (2). These holes (h^+^) can strongly oxidize MB (Equation (3)). On the other hand, e^−^ can react with oxygen (dissolved in the reaction mixture) and can generate superoxide radicals (·O_2_^−^), as given in Equation (4). According to Equation (5), the photoexcited electrons (e^−^) can migrate from the MOF to BiOI, which inhibits electron–hole pair recombination and enhances MB degradation under visible-light illumination [18,47]. The superoxide radicals (·O_2_^−^) finally degrade MB into degradation products, that is, CO_2_ and H_2_O, as shown in Equation (6): A schematic representation of this process is given in Appendix A.

### 4.10. Recyclability of Photocatalyst

The recyclability of a photocatalyst is a key factor for evaluating its effectiveness in industrial applications. To examine the recyclability of the prepared photocatalyst, Co/Ni-MOF@BiOI-10 was recovered by centrifuging the reaction mixture after every MB degradation cycle. Before the next MB degradation run (using the optimized conditions of pH, catalyst amount, and MB concentration), the photocatalyst was washed with ethanol and dried under vacuum. As shown in Figure 8, the photocatalytic efficiency of Co/Ni-MOF@BiOI-10 did not change after ten successive cycles. This recycling capacity further supports the potential of Co/Ni- MOF@BiOI-10 as an efficient photocatalyst.

## 5. Conclusions

Co/Ni-MOF@BiOI nanocomposites were successfully synthesized using a solvothermal method and characterized using various techniques. Subsequently, the photocatalytic activities of the prepared catalysts toward MB degradation under visible light were investigated. Among all the catalysts, the maximum photocatalytic efficiency of approximately 99% was observed for Co/Ni-MOF@BiOI-10 composite when compared to Ni-MOF, Co/Ni-MOF, and Co/Ni-MOF@BiOI-5 without adding any chemical additives. The enhanced performance of Co/Ni-MOF@BiOI-10 was attributed primarily to the addition of BiOI, which reduced electron–hole pair recombination and expanded the light absorption to the visible region. Further investigation of the effects of pH, catalyst dose, and MB concentration revealed that the photocatalytic efficiency of the as-prepared photocatalyst increased with increasing catalyst dosage and under slightly acidic conditions, and that a pH value of approximately 5 was more suitable for the degradation of MB. Based on these results, it can be concluded that this novel combination of MOF/BiOI prepared in situ may serve as a potential photocatalyst for the degradation of organic contaminants.

## Figures and Tables

**Figure 1 micromachines-14-00899-f001:**
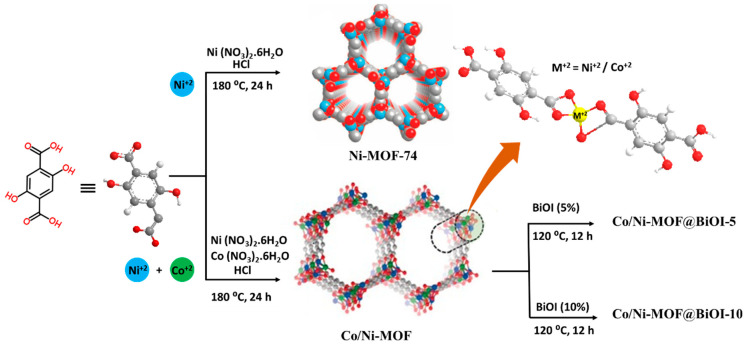
Synthesis of Co/Ni-MOF@BiOI composites.

**Figure 2 micromachines-14-00899-f002:**
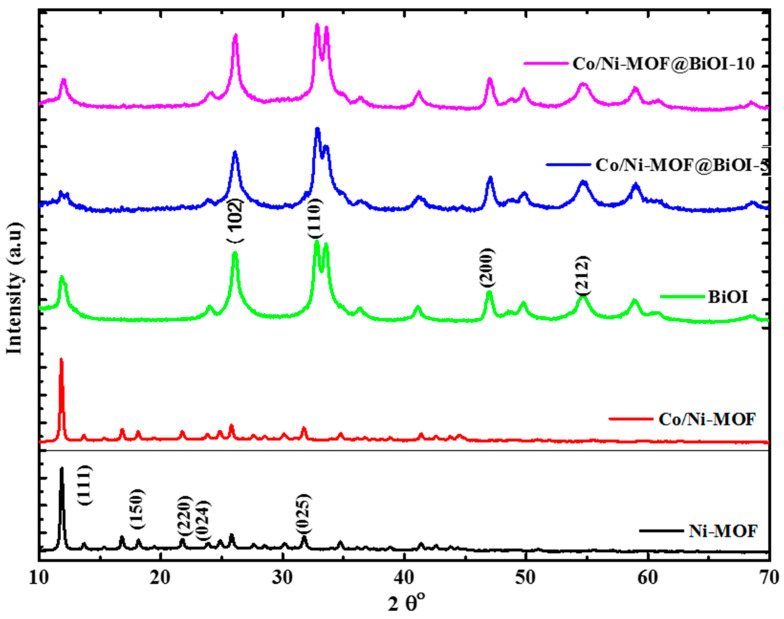
XRD patterns of Ni-MOF, Co/Ni-MOF, BiOI, CO/Ni-MOF@BiOI-5, and CO/Ni-MOF@BiOI-10.

**Figure 3 micromachines-14-00899-f003:**
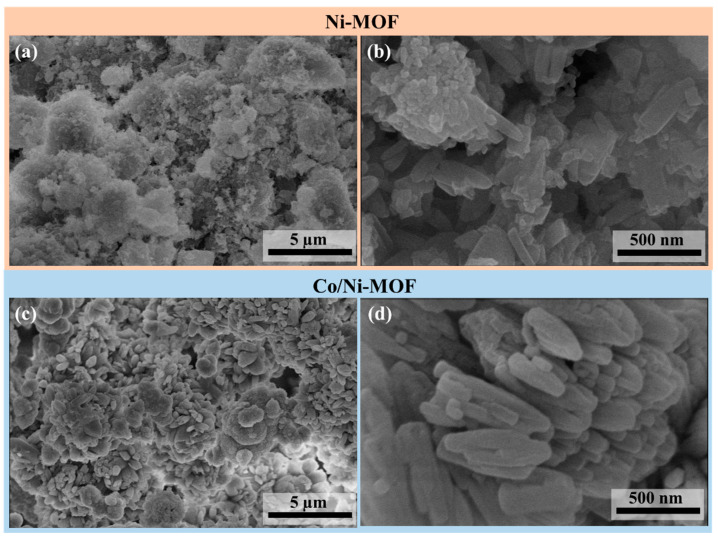
SEM micrographs of (**a**,**b**) Ni-MOF and (**c**,**d**) Co/Ni-MOF.

**Figure 4 micromachines-14-00899-f004:**
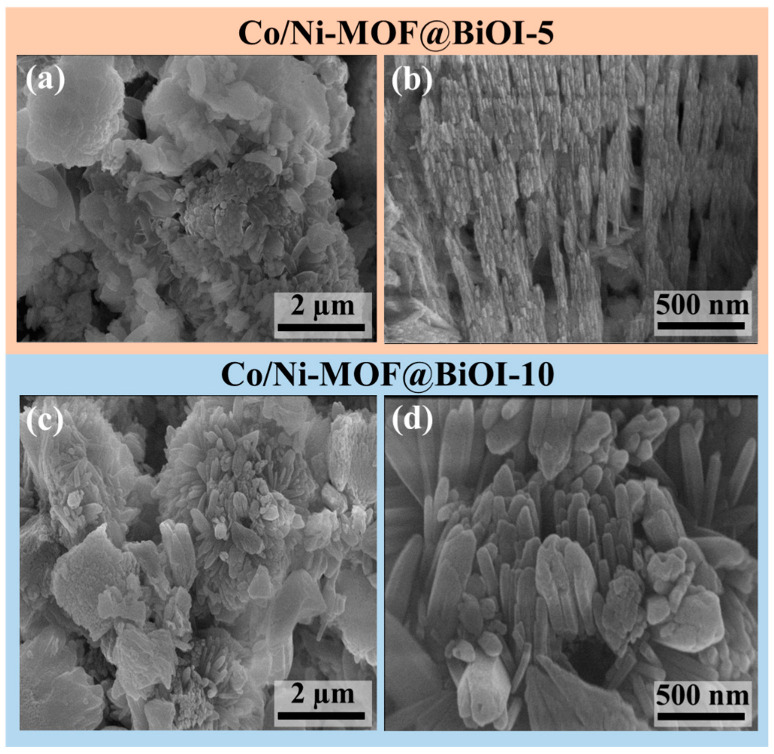
SEM images of (**a**,**b**) Co/Ni-MOF@BiOI-5 and (**c**,**d**) Co/Ni-MOF@BiOI-10.

**Figure 5 micromachines-14-00899-f005:**
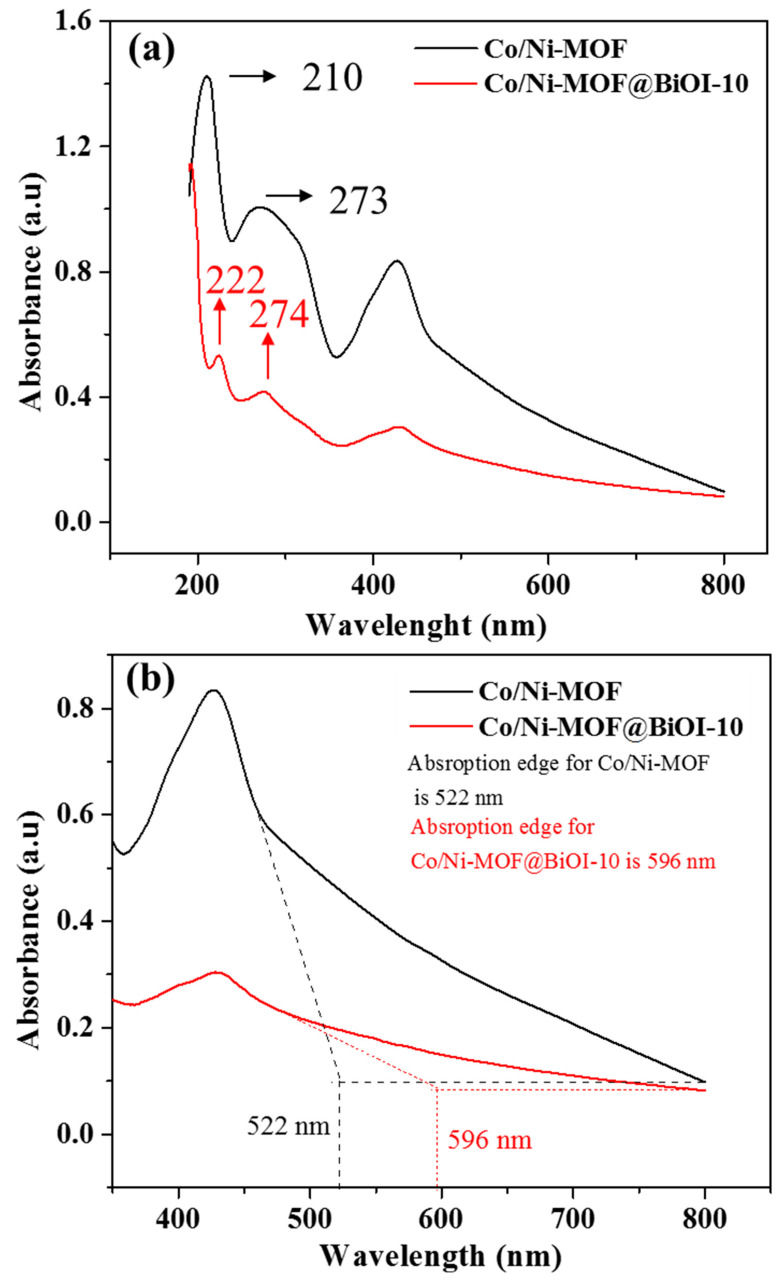
(**a**) UV–vis spectra of Co/Ni-MOF and Co/Ni-MOF@BiOI-10. (**b**) UV–vis spectra of Co/Ni-MOF and Co/Ni-MOF@BiOI-10 showing visible range of spectrum and highlighted absorption edges of the both samples.

**Figure 6 micromachines-14-00899-f006:**
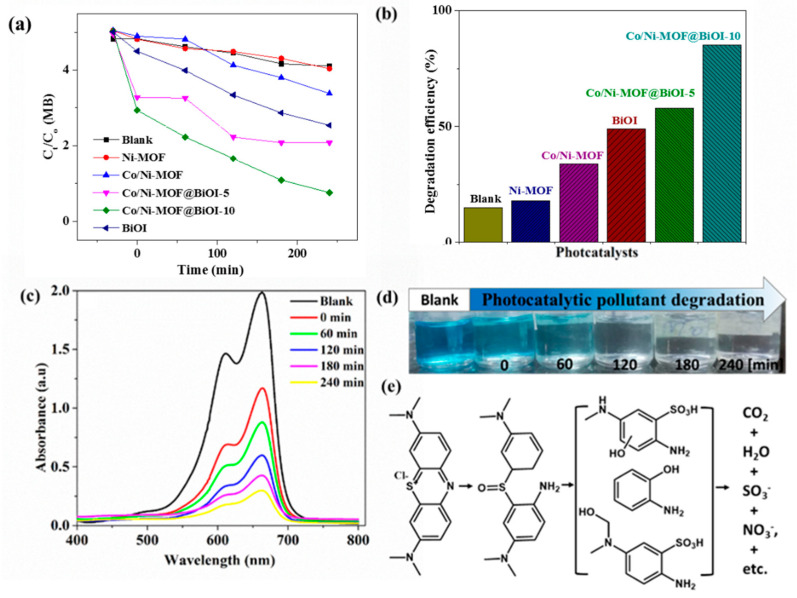
(**a**) MB concentration as a function of exposure time under sunlight. (**b**) Efficiency of catalysts toward MB degradation. (**c**) UV–visible spectra of MB (in aqueous solution) at different irradiation time intervals in the presence of Co/Ni-MOF@BiOI-10. (**d**) Photographs of the MB solutions in the presence of Co/Ni-MOF@BiOI-10 over time under sunlight. (**e**) Suggested photocatalytic degradation pathway of MB molecules reacting with Co/Ni-MOF@BiOI-10.

**Figure 7 micromachines-14-00899-f007:**
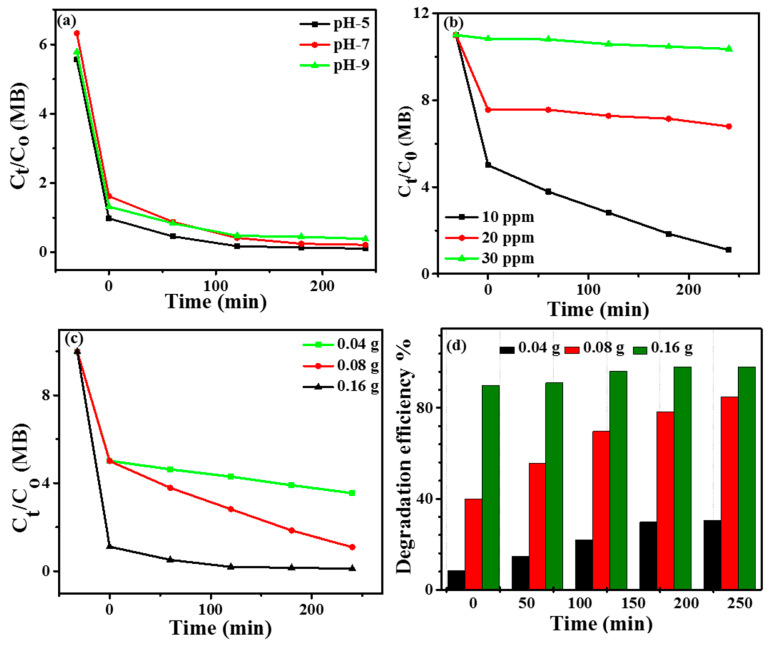
(**a**) The effect of pH on the catalytic efficiency of Co/Ni-MOF@BiOI-10; (**b**) the effect of initial concentration of MB; (**c**) the influence of catalyst amount on the photocatalytic activity of Co/Ni-MOF@BiOI-10; and (**d**) the degradation efficiency of MB versus irradiation time at different catalyst dosage.

**Figure 8 micromachines-14-00899-f008:**
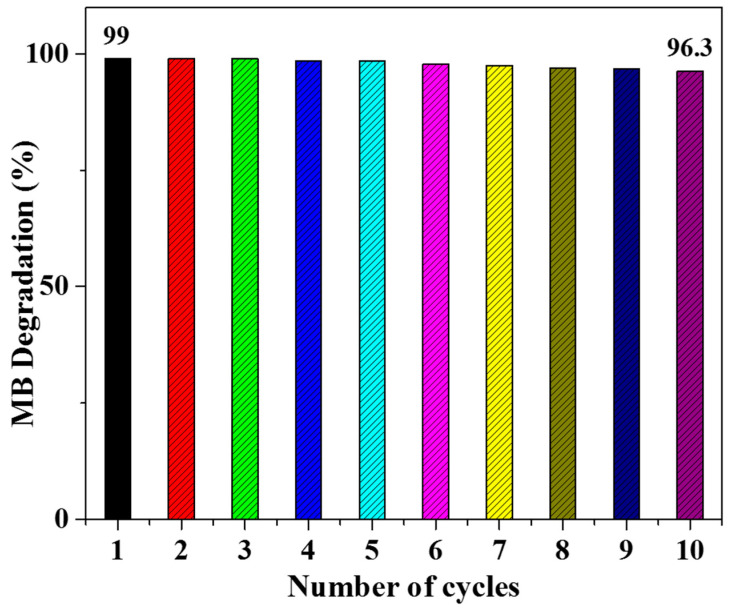
Removal efficiency (%) versus recycle times of Co/Ni-MOF@BiOI-10 for MB degradation.

**Table 1 micromachines-14-00899-t001:** Photocatalytic performance of various catalysts applied for degradation of MB.

Photocatalysts	Time (min)	MB Degradation (%)	References
Fe_2_TiO_5_ nanoparticles	240	~53	Ref [42] of sana
Ag_3_PO_4_/MnFe_2_O_4_ NCs	82	98	[43]
AC-Bi/TiO_2_ NCs	100	97	[44]
MOF-5@rGO	20	93	[45]
SnO_2_/TiO_2_ nanocomposites	50	~90	[46]
Ag.Co_3_O_4_ nanosheets	90	88.4	[47]
Ni/Cu-BDC MOF@BiOI-15%	240	99	This work

## Data Availability

Not applicable.

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
