# Peer review of "Bismuth-Rich Co/Ni Bimetallic Metal–Organic Frameworks as Photocatalysts toward Efficient Removal of Organic Contaminants under Environmental Conditions"

_micromachines, 2023, doi:10.3390/mi14050899_

Round 1

Reviewer 1 Report

1.      The experimental procedure is hard to understand and should present precisely and clearly to improve readability.

2.      The authors state that the molar ratio of as-prepared Co/Ni-MOF-74 was Co/Ni = 0.5%, but how they calculated the composite's Ni/Co molar ratio needs to be clarified.

3.      The authors should provide data for ICP and CHNS elemental analysis and assign the atomic molar ratios accordingly.

4.      Thorough characterization using ICP and CHNS is required for all synthesized materials (Ni-MOF-74, Co/Ni-MOF-74, Co/Ni-MOF@BiOI-5, and Co/Ni-MOF@BiOI-10).

5.      The authors should perform Rietveld refinement on the composite Co/Ni-MOF-74 to support their claim of volume expansion of the crystal lattice from 0.42 nm to 0.46 nm and compare atomic coordinates between Ni-MOF-74 and Co/Ni-MOF-74.

6.      Powder-XRD of BiOI, Co/Ni-MOF@BiOI-5, and Co/Ni-MOF@BiOI-10 show similar signals, but no apparent signals were observed for MOF-74. The authors need to explain why the composite does not offer any parent signals for MOF-74, despite the composite having just 5 to 10% BiOI concentration.

7.      The authors observe that both Co/Ni-MOF@BiOI-5 and Co/Ni-MOF@BiOI-10 maintain the parent MOF rod-like morphology, but both PXRD and SEM are contradictory. The authors should provide a better explanation and use HR-TEM to understand it better.

8.      The authors must clearly explain Figures 5a, 5b, and 5c and remove unnecessary figures. They should also improve the image quality for publication.

9.      The authors need to present the photocatalytic activity and MB degradation performance of BiOI alone, along with other materials, in the manuscript.

10.   The authors should compare the photocatalytic activity and MB degradation of all presented materials (Ni-MOF-74, BiOI, Co/Ni-MOF@BiOI-5, and Co/Ni-MOF@BiOI-10) with literature values in a separate table.

11.   The authors mention that the pore size and specific surface area were characterized using Brunauer-Emmett-Teller (BET), but no data has been presented.

12.   The effect of various parameters, such as pH, concentration, catalyst dosage, and recyclability, is interesting but insufficient for publication in Micromachines.

Author Response

Please check out the enveloped letter

Reviewer 2 Report

Report on the manuscript micromachines-2336619 entitled “Bismuth-Rich Co/Ni Bimetallic Metal-Organic Frameworks as Photocatalysts toward Efficient Removal of Organic Contaminants in Environmental Condition”.

The submitted review should be revised. The following points should be addressed:

1. The submitted manuscript should be revised to be free from editing or grammar errors. For example, “obtained precipitate” should be “the obtained precipitate”, …etc.

2. In experimental work, the role of HCl should be supported and it should be indicated in scheme 1.

3. How the ratio of Co/Ni = 0.5 % and the amount of Co-salt is around 50 % of Ni-salt.

4. Equation1 should be adjusted in editing, the 0 and t should be subscript and same for figure 7A.

5. In XRD, the corresponding to plane (11-1)”, please, correct it. Try to support the JCPDS number and crystal size analysis.

6. According to SEM images, the Ni-MOF has micro-size not nanoscale?, so, how the authors estimate diameter of nanorods to be 48.5 nm.

7.  In UV-visible spectroscopy, there is clear peak in the earlier region and disappeared after Co-modification, why?

8. “therefore, they didn’t exhibited any significant photocatalytic activity under visible light.”, the visible region started from 400 nm why Ni-MOF and Co/Ni-MOF will be inactive in visible region?

9. What about adding more amount from catalysts as 0.16 is the highest and best so, maybe more will be better?

10. 3 cycles are limited, please, try to extend these to 10 cycles for recyclability study?

11. [Important] Isotherm study and mechanistic aspects should be studied as this is the basis of photocatalytic study.

12. The authors couldn’t confirm the chemistry of the prepared materials, XPS or/and EDX should be supported.

Author Response

Please check out the enveloped letter

Round 2

Reviewer 1 Report

Dear Authors,

Thank you for incorporating the suggested data. The current manuscript has shown significant improvement from the first submission.

Reviewer 2 Report

The revised version could be accepted.